# Resistance Mechanism of *Plutella xylostella* (L.) Associated with Amino Acid Substitutions in Acetylcholinesterase-1: Insights from Homology Modeling, Docking and Molecular Dynamic Simulation

**DOI:** 10.3390/insects15030144

**Published:** 2024-02-21

**Authors:** Maryam Zolfaghari, Yong Xiao, Fardous Mohammad Safiul Azam, Fei Yin, Zheng-Ke Peng, Zhen-Yu Li

**Affiliations:** 1Plant Protection Research Institute, Guangdong Academy of Agricultural Sciences, Guangzhou 510640, China; m.zolfaghari_89@yahoo.com (M.Z.); xiaoyong@gdaas.cn (Y.X.); feier0808@163.com (F.Y.); zkpeng0827@163.com (Z.-K.P.); 2Guangdong Provincial Key Laboratory of High Technology for Plant Protection, Guangzhou 510642, China; 3College of Life Science, Neijiang Normal University, Neijiang 641100, China; shojibbiotech@yahoo.com; 4Department of Biotechnology and Genetic Engineering, Faculty of Life Sciences, University of Development Alternative, Dhanmondi, Dhaka 1209, Bangladesh

**Keywords:** point mutation, insecticide resistance, molecular dynamics simulation, gene expression, diamondback moth

## Abstract

**Simple Summary:**

One of the most destructive pests of cruciferous plants that quickly develops resistance to most pesticide groups is *Plutella xylostella*. Investigating the resistant strain of *P. xylostella* revealed the molecular mechanisms underlying resistance to chlorpyrifos, concentrating specifically on the *ace1* gene. The sequencing results revealed amino acid substitutions in *ace1* of the resistant strain. The structures of the wild-type and mutant *ace1* strains were compared via molecular dynamics (MDs) simulations and docking investigations. The results showed that the mutant *ace1* has different substrate entry points and structural modifications that affect the enzyme inhibitor affinity. Significant differences in *ace1* gene expression between the mutant and wild-type strains were revealed by real-time quantitative PCR, which raises the possibility of a relationship between *ace1* mutations and changes in mRNA transcription levels.

**Abstract:**

*Plutella xylostella,* a destructive crucifer pest, can rapidly develop resistance to most classes of pesticides. This study investigated the molecular resistance mechanisms to chlorpyrifos, an organophosphate pesticide. Two *P. xylostella* genes, *ace1* and *ace2*, were described. The nucleotide sequence results revealed no variation in *ace2,* while the resistant strain (Kar-R) had four amino acid alterations in *ace1*, two of which (A298S and G324A) were previously shown to confer organophosphate resistance in *P. xylostella.* In the present study, the 3D model structures of both the wild-type (Gu-S) and mutant (Kar-R) of *P. xylostella ace1* strains were studied through molecular dynamics (MDs) simulations and molecular docking. Molecular dynamics simulations of RMSD revealed less structural deviation in the *ace1* mutant than in its wild-type counterpart. Higher flexibility in the 425–440 amino acid region in the mutant active site (Glu422 and Acyl pocket) increased the active site’s entropy, reducing the enzyme’s affinity for the inhibitors. Gene expression analysis revealed that the relative transcription levels of *ace1* were significantly different in the Kar-R strain compared with the Gu-S strain. This study enhances the understanding of the mechanisms governing *ace1*′s resistance to insecticide and provides essential insights for new insecticides as well as valuable insights into environmentally conscious pest management techniques.

## 1. Introduction

*Plutella xylostella* L., a widespread lepidopteran insect pest known as the diamondback moth (DBM) (Lepidoptera: Plutellidae), infests a broad range of vegetables and crops all over the world. It is estimated that USD 4 to 5 billion are needed annually to eradicate and manage this pest [1,2]. To efficiently control *P. xylostella* and satisfy people who cultivate cruciferous vegetables, insecticides are primarily utilized against the pest. However, overusing pesticides to suppress *P. xylostella* may result in insecticide resistance across all classes of pesticides [3,4].

Organophosphates (OPs) were introduced several decades ago as effective pest control agents and account for more than 30% of the registered synthetic insecticides and acaricides in the United States [5]. Organophosphate pesticides act by inhibiting the acetylcholinesterase enzyme (AChE). This enzyme alters the acetylcholine (ACh) level in the synaptic cleft. It catalyzes the hydrolysis of the neurotransmitter ACh to acetic acid and choline in synapses and neuromuscular junctions in both vertebrates and invertebrates [6]. Research on AChE, an important enzyme that organophosphorus (OP) and carbamate (CB) pesticides target, has been extensive, throughout a wide range of insect pest species. One or two *ace* genes encode this enzyme. AChE1 is significantly more highly expressed than AChE2. The majority of insects carry both *ace* alleles. The two AChEs have completely distinct substrate and inhibitor preferences. This issue was first identified in two distinct *Culex pipiens* loci, and it was subsequently later determined that AChE was intended to be encoded by two paralogous genes [7]. However, several insect species, such as *P. xylostella* [8], *Myzus persicae* [9], *Culex* spp. [10,11] and *Spodoptera frugiperda* [12], have two *aces*.

The acyl pocket comprises three residues (i.e., Phe384, Phe427, and Trp330), while the catalytic triad consists of His 537, Ser 323, and Glu 423. The choline-binding site is comprised of just one residue, Trp182. According to [13], these catalytic residues play a crucial role in regulating the binding orientation. One of the primary enzymes targeted by pesticide formulations is AChE1, and many of these formulations, especially organophosphates, have been created to suppress this enzyme [14]. When an insecticide molecule is positioned in the active site, as in the case of chlorpyrifos, the phosphate group often forms covalent adducts with the reactive serine 200 hydroxyl group. Consequently, it inhibits the hydrolase from binding acetylcholine, overstimulating cholinergic synapses and killing insects [15].

AChE insensitivity to CB and OP insecticides results from structural alterations in the *ace1* subtype of AChE. These changes are the result of amino acid substitution [8]. Over 33 insects and Acari species [16], including *Leptinotarsa decemlineata* [17], *Bactrocera oleae* [18], *Culex* spp. [10], *M. persicae* [9], and *S. frugiperda* [12], have evolved resistance to CB and OP insecticides due to decreased AChE sensitivity.

The molecular structure of insensitive AChE has recently been extensively studied in several insects. For instance, four amino acid substitutions (A201S, F290V, G227A, and AF290V) in *S. frugiperda’ ace1* [12], and five alterations (V180L, G262A, G262V, F327Y, and G365Y) were found in OP-resistant house fly strains [19]. The fruit fly contains the following five mutations that confer insecticide resistance: F115S, I119V, I119T, G303A, and F368Y [20]. Additionally, three mutations (D132G, A201S, and G227A), that cause prothiofos resistance, were found in *P. xylostella*. Of the two mutations of this pest (A201S and G227A), A201S is close enough to the enzyme’s catalytic activity. At the same time, G227A may impact the enzyme’s activity by changing the area around the active site [13].

One of the pesticides frequently used in Iran is chlorpyrifos, due to its low cost and range of effects. This insecticide is frequently and extensively used to manage a variety of pests, including *P. xylostella*. This study builds on the findings of [21] regarding a chlorpyrifos-resistant *P. xylostella* population in Karaj, Iran. Thus, in this present experimental study, we cloned and sequenced the *ace1* and *ace2* genes and identified four mutations (A298S, G324A, F470L, and T473I) in the *ace1* Kar-R strain. The impact of mutations in OPs binding to *ace1* was estimated through molecular docking (MD) and gene expression analyses. Behavioral changes in the protein were used to predict the three-dimensional structure of AChE1. All-atom MD simulations were carried out to determine the influence of mutations on the structural conformation and residual flexibility [22].

## 2. Materials and Methods

### 2.1. Insects

A population sensitive to chlorpyrifos (GU-S) was gathered from a cabbage field in Rasht, Guilan Province, Iran, and reared in the laboratory under controlled conditions (25 ± 2 °C, 65 ± 10% RH, and 16:8 (L:D) h photoperiod) on *Brassica rapa* for three years without exposure to any pesticides. The strains resistant to chlorpyrifos were found in Karaj, Alborz Province, Iran (Kar-R). Five repetitions of the bioassay experiments were conducted, with ten-thirds of instar larvae of the same age being utilized in each replication. The result of the bioassay from our previous research [21] indicated (LC_50_ = 62 g/mL) for the susceptible population and (LC_50_ = 4300 g/mL) for the resistant population.

### 2.2. RNA Extraction, cDNA Synthesis, Cloning and Sequencing

To induce resistance, the resistant strain was treated with the highest dose of chlorpyrifos (9000 μg/mL) through a leaf-dipping test as outlined by [20]. At 4 d post-treatment, the treated *P. xylostella* pests were collected. Larvae were homogenized by a mortar and pestle in liquid nitrogen. Using 1 mL of RNX reagent (SinaClon, Tehran, Iran), 3 larvae for each of two resistant (Kar-R) and susceptible (Gu-S) strains were subjected to total RNA extraction according to the manufacturer’s instructions. The RNA concentration was checked by a NanoDrop™ One microvolume UV-Vis spectrophotometer (Thermo Scientific, Wilmington, DE, USA) based on the absorbance ratio of OD260/280. Furthermore, the RNA integrity was confirmed by 1% agarose gel electrophoresis. To remove genomic DNA from the RNA, l μL of DNase H (Invitrogen Co., San Diego, CA, USA) was added for 20 min at 37 °C. Using oligo (dT) as a primer, first-strand complementary cDNA was produced from total RNA using a cDNA synthesis kit (Takara, Shiga, Japan).

Next, unique primers (Table 1) were designed from the GenBank database using Gene Runner software ver. 6.5.52 (www.generunner.net; accessed on 8 February 2024) to amplify cDNA fragments of *P. xylostella* (*ace1* and *ace2*) from the first-strand cDNA [12].

PCRs were conducted using a DNA Engine Dyad Peltier Thermal Cycler from Biorad in Hercules, California, and EX Taq Polymerase from Takara in Shiga, Japan. According to the following program, PCRs were conducted: 35 cycles at 94 °C for 40 s, 62 °C for 30 s, and 72 °C for 120 s, with the last extension lasting for 5 min at 72 °C. The PCR products were extracted via gel electrophoresis using Qiagen’s Qiaquick gel extraction kit (Valencia, CA, USA).

The instruction manual of the TOPO TA Cloning Kit (Invitrogen) was followed and then 60 ng/l of pure DNA fragments were cloned and inserted into the PGEM^®^-T vector. Using PCR amplification with the M13 and T7 primers, transformants (white colonies on LB plates treated with ampicillin) were evaluated for insertions. A single white colony, 7.5 μL of Master mix 10× (Takara), 0.5 μL of each primer (M13 and T7) containing ten pmol, and 15 μL of sterile filtered water were used as the components of the PCR mixture. PCR was performed following the aforementioned protocol. Using a Qiagen Kit (Qiagen, Hilden, Germany), plasmid DNA was extracted from the positive transformants. Before sequencing, the M13/T7 primers were used to prepare duplex DNA from the isolated plasmid. Three clones were sequenced for each strain using an ABI Model 3100 automated sequencer (Invitrogen Life Technologies, Shanghai, China) to verify allele representations. GeneDoc software v2.7 (https://genedoc.software.informer.com/2.7/; accessed on 8 February 2024) was used for DNA sequence analysis. The sequences were compared using Clustal Omega software online (https://Tools/msa/clustalo; accessed on 12 May 2020)

### 2.3. qRT-PCR Assay

Total RNA was extracted from third-instar *P. xylostella* larvae using TRIzol. We measured the expression of the *ace1* and *ace2* genes via qRT-PCR which was carried out using the Maxima SYBR Green Rox qPCR master mix kit (Fermentas, Burlington, ON, USA).

A total of three biological replicates were used for each treatment. The primers used were synthesized by Invitrogen Trading (Shanghai) Corporation Ltd. The details of the primers used are shown in Table 1. Briefly, 0.5 μL of primers (forward and reverse) (10 mM), 7.5 μL of SYBR Green PCR Master Mix, 1 μL of cDNA, and 3 μL of nuclease-free water from control and S and R population larvae were added to the qPCR mixture in a total volume of 12.5 μL. The following thermal cycling procedure was used: 2 min at 95 °C, 30 s at 95 °C, 30 s at 57 °C, 30 s at 72 °C, and 5 min at 72 °C. The 2^−∆∆Ct^ approach was applied to compute the relative quantification of gene expression [23].

### 2.4. Computational Study

#### 2.4.1. Structural Modeling

In this study, we utilized the SWISS-MODEL online platform (https://swissmodel.expasy.org/; accessed on 10 June 2020) to construct models for two protein sequences (Kar-R and Gu-S) [24]. Choosing the model with higher coverage, we further validated it on SAVES v6.0 (UCLA-DOE LAB, (Los Angeles, CA, USA)) for model accuracy (https://saves.mbi.ucla.edu/; accessed on 21 June 2020).

From the PubChem database, we identified the compound name chlorpyrifos oxan (CAS-No. 2921-88-2), which has a molecular weight of 350.6 g/mol and a 3D structure. The corresponding structure data file (SDF) was downloaded, and ligands and proteins were prepared using AutoDock Vina v.1.1.2 software for molecular docking [25]. The crystal structure of the target protein underwent preprocessing, ensuring low-energy conformations for ligand structures. The docking of the target structure with the active compound structure was performed using the Vina tool within PyRx 0.8 software (http://pyrx.sourceforge.net; accessed on 8 February 2024). The binding strength between the ligand and receptor was represented by the affinity (kcal/mol) value, where lower values indicated more stable binding. To analyze and visualize the docking results, BIOVIA Discovery Studio Visualizer 2020 (https://discover.3ds.com; accessed on 8 February 2024) and PyMOL were used [26].

#### 2.4.2. Structure Validation

The protein structure predictions were subjected to scrutiny through a Ramachandran plot to assess their rationality. These plots were also created to show the dihedral angles ψ and φ of the protein’s main chain amino acid residues. The rotation angles in this plot represent the left C-N bond (φ) of the right C-C bond (ψ) of the α-carbon. According to [27], the Ramachandran plot helps assess the structural integrity of protein conformations. According to [28], the Ramachandran plot shows the percentages of amino acids in different categories, including the most favored region, additional allowed region, generously allowed region, and disallowed region.

#### 2.4.3. Molecular Simulation

The crystal structure of acetylcholinesterase (AChE) is unavailable and thus we employed molecular modeling to construct the theoretical model. The AChE amino acid sequence was obtained from UniProtKB (accession number A0A8S4E7J2, 825 amino acid residues), and the protein model was constructed using the I-TASSER server which threads the protein sequence from the PDB structure library and searches for possible alignments using LOMETS [29,30], incorporating ab initio modeling for unaligned regions. PROCHECK was subsequently used to determine the consistency and stereochemical properties of the model [31].

In addition, the wild-type A298S, G324A, F470L, and T473I mutations were induced to generate the R mutant (resistant). The desired mutations were applied to the relevant sites, and repeated cycles of energy minimization were performed using Spdbv software (version 4.0.1). Then, molecular dynamics simulations were carried out using the GROMACS v5.1.2 package [32] with the force fields of the Amber f99SB [33] and TIP4P water models. The proteins were simulated in a water-filled box, the distance was 1.2 nm between the protein and the box, and sodium ions were added to neutralize the system. In all three directions of space, periodic boundary conditions were applied. The steepest descent algorithm and the conjugate gradient method were used in the systems to reduce for 1000 steps to remove any unfavorable interaction between atoms. Then, the system was gradually heated to 300 K by a 50 ps NVT simulation and equilibrated by a 500 ps NPT simulation at 1 atm. These processes were performed at a constant temperature (300 K, Langevin thermostat) and pressure (1 bar, Berendsen barostat) with a relaxation time of 2 fs [34]. Next, a 10 ns unrestrained production simulation was conducted for the system using an integration time step of 2.0 fs. The entire simulation procedure was conducted with a 10 Å cutoff for nonbonded interactions. Finally, the structural difference between the two protein conformers was expressed in terms of the root mean squared deviation (RMSD) to confirm that, after the initial 10 ns, both systems were well equilibrated. Additionally, the root mean squared fluctuation calculated the fluctuations in the residues of Ca atoms.

### 2.5. Data Analysis

Using SAS version 9.2 software (SAS Institute, 2004), the mean values of the recorded data were analyzed, followed by a *t*-test where differences were deemed significant at *p* < 0.05. GMX solvate command within GROMACS 2021.5 software were used to conduct molecular dynamics simulations, and Antechamber was used to evaluate the parameters for the force field of the molecules that were inhibited [35].

## 3. Results

### 3.1. Gene Sequencing Revealed Amino Acid Substitutions in the ace1

By PCR amplification and cDNA sequencing of the two populations, the *P. xylostella ace1* and *ace2* genes from the Gu-S and Kar-R strains were further confirmed and the results using Bioinformatics Sequence Translation tools were translated to amino acids (https://web.expasy.org/translate/; accessed on 26 May 2020) (Figure 1). Multiple sequence alignment revealed that both *ace1* and *ace2* have conserved AChE motifs, including choline-binding sites, a catalytic triad, an acyl pocket, an oxyanion hole and a peripheral anionic subsite. The functional motifs of *P. xylostella* in *ace1* were markedly preserved compared to those in *ace* in *Torpedo californica* (Figure 2). The *P. xylostella ace1* gene is encoded by a 1326-bp fragment that contains 596 amino acid residues. Three cDNA clones from each population were sequenced throughout this procedure. Sequence analysis revealed variations in the amino acid sequences of the strains and mutations in the resistant strain. Four amino acid position alterations were found in the Kar-R strain, with the numbers matching those of the adult enzyme of *Torpedo californica* (Torpediniformes: Torpedinidae): A298S (TCC-to-GCC substitution), G324A (GGA-to-GCA substitution), T473I (ACT-to-ATT substitution), and F470L (TTC-to-TTG substitution) (Figure 2). Additionally, the cDNA coding regions of both the susceptible and resistant strains of *ace2* encoded 1387 bp segments and 462 residues. The findings revealed no differences when the *ace2* gene snippets overlapped ([8]).

### 3.2. Significant Differences in ace1 and ace2 Gene Expression

Real-time quantitative PCR was performed to ascertain the mRNA transcription levels of *ace1* and *ace2* in the Gu-S and Kar-R strains. The transcription level of *ace1* mRNA in the Kar-R strain was lower than that in the Gu-S strain, and there was no discernible difference in the transcription levels of *ace2* in the susceptible and resistant strains (Figure 3). The *p* value for *ace1* was 0.0392, whereas for *ace2*, it was 0.86. The resistant strain (Kar-R) exhibited a substantial 90% reduction in the *ace2* expression compared to the susceptible strain (Gu-S).

### 3.3. Molecular Dynamics Simulations

#### 3.3.1. Amino Acid Substitutions Altered Binding Energy between Mutant Protein and Ligand

As shown in the illustration (Figure 4A), in the susceptible protein (Gu-S), the small molecule chlorpyriphos oxan (ligand) (CAS-No. 2921-88-2) interacts with the AChE amino acids. The binding energy is −6.0 kcal/mol, indicating that these two molecules have strong interactions. Figure 4B depicts the interaction between the S-protein and the ligand. In Figure 4C, there is an interaction between the mutant protein (Kar-R) and the ligand (CAS-No. 2921-88-2), where the binding energy is −5.9 kcal/mol. This indicates that these two molecules can interact; however, their affinity is lower, and they require less energy to break the interaction state between the ligand and protein of the resistant strain than the susceptible one. Figure 4D depicts the interaction between the mutant protein and the ligand (CAS-No. 2921-88-2). For instance, “ILE(A:129)” denotes the presence of the amino acid ILE at position 129 on chain A, signifying the interaction between the ligand and the amino acid ILE positioned at 129 surrounding the R-protein. Moreover, this interaction lost the binding site at TYR A: 392 of the protein with the pesticide.

#### 3.3.2. Ramachandran Plots Validated the Structural Models

When evaluating models for homologous proteins, it is typically expected that the percentage of amino acids situated in the disallowed regions is less than 5% of the total amino acid count. Notably, the percentage of amino acid residues in disallowed regions was 0%, underscoring the reasonableness of the modeling for both susceptible (Figure 5A) and resistant (Figure 5B) strains. Consequently, the modeled structure was deemed suitable for further docking analysis.

#### 3.3.3. Higher Flexibility and Lower Affinity of Resistant AChE for Chlorpyriphos

The root-mean-squared deviation (RMSD) between generated structures in the simulation serves as an appropriate optimality criterion for ensuring system equilibration and extracting information about possible severe conformational changes in the protein. Therefore, the RMSD changes for the alpha-carbon atoms of two proteins were calculated and extracted at different temperatures during the simulation (10 ns), compared to the original structure (Figure 6). The range of RMSD variations for both protein variants is depicted in Figure 6. In the R protein structure, the *ace1* enzyme stabilized after approximately 100 picoseconds (ps) of simulation and exhibited an RMSD of 0.1 nm with slight and low fluctuations toward the end of the simulation. The sensitive (S) *ace1* enzyme displayed a linear increase in RMSD with a steeper slope, reaching a plateau after 700 ps, whereas the RMSD of the R variant remained unchanged after 300 ps. The stability of the trajectories in the simulation was confirmed by the minimal differences between the average RMSD values after 6000 ps, and these fluctuations revealed the feasibility of binding and interactions.

The fluctuations of the residues are analyzed by calculating the root mean squared fluctuation (RMSF) of the backbone atoms in the last 2 ns of the molecular dynamics simulation (Figure 7). The dynamic behavior of alpha carbon atoms in a structure requires sufficient information to investigate necessary protein motions and reflects general structural movements. The most stable and minimal fluctuations are shown when the residues interact with ligands. Therefore, the RMSF was used to investigate the effects of mutations on fluctuations in neighboring amino acids.

In this section, the structural flexibility of the Kar-R and Gu-S *ace1* enzymes was compared, as illustrated in Figure 8. Figure 8 shows the two proteins exhibit similar flexibility in most regions. However, at some points, there are also significant differences. For example, amino acids 580 to 700 in the susceptible *ace1* (S green) exhibit a high degree of flexibility compared to the resistant protein, which may be a reason for the high RMSD of this enzyme during the simulation period compared with the other enzymes. There is a flexible region (425–440) in the resistant protein that is located in the active site of the enzyme; thus, it can be concluded that increased flexibility in this region can increase the entropy (ΔS) in the active site of the protein by altering the size of the inhibitor inlet gap. This effect can decrease the inhibitor’s affinity for the enzyme. Increasing the flexibility of this region can also alter the size of the inhibitor inlet gap, thereby reducing the tendency of the enzyme to bind to the inhibitor by altering the interaction.

#### 3.3.4. Modifying the Active Site of Resistant AChE Protein through Amino Acid Substitutions

The expected model for binding sites must be identified for docking research. Because the resistant protein structure is unavailable in the database and the binding site for pesticides must be predicted, the interaction site with the ligand binding pocket was chosen to dock the chlorpyriphos oxan. Once 3D models of the wild-type (Kar-R) *ace1* enzyme were constructed, there was appreciable variation in the active site compared to that of the wild-type (Gu-S) *ace1* enzyme (Figure 9). The chlorpyriphos oxan molecule, in complex with the *ace1* mutant of *P. xylostella*, presented a different binding pattern than that of the wild-type; however, it occupied the same binding pocket. In the mutant protein, the above organophosphates cannot interact at the active site due to four substitutions in amino acids found in the mutant protein. The acid position substitution sites were A298S, G324A, F470L, and T473I, which yielded amino acid changes from alanine to serine, glycine to alanine, phenylalanine to leucine, and threonine isoleucine, respectively, in the resistant protein. However, the mutant has a three-dimensional change in the active site and is unable to interact with the pesticide.

## 4. Discussion

As a crucial enzyme for cholinergic nerve transmission, insect AChE is the target of pesticides made of carbamates and organophosphates [36]. Point mutations in *ace* genes are often impacted by insects’ resistance to pesticides [37]. As a result, in recent years, these genes have received much interest in the area of pesticide toxicity [38,39]. As has been reported at the molecular level in most instances to date, the insect’s *ace1* gene is involved in target site resistance due to mutations [40,41]. Several studies have shown no connection between the *ace2* gene and OP resistance [42]. However, there have been unusual occurrences, such as the correlation between OP resistance and mutations in both *ace1* and *ace2,* which are found in the wheat aphid, *Sitobion avenae* [40,43].

Understanding the significance of mutation sites in genes that generate pesticide resistance is crucial for developing sustainable pest management strategies. Pesticide resistance often involves mutations in specific genes related to the target site of the pesticide. Understanding specific mutation sites in the target genes helps predict the effectiveness of different pesticides. The significance of mutation sites lies in their role in the evolutionary dynamics of resistance. Mutations may confer resistance but often come with fitness costs, and understanding these trade-offs is essential for sustainable pest management [44,45].

There are many reasons why different resistant strains have different mutation locations. First of all, different mutations may result from variations in pesticide use and selection pressures in various geographic locations or agricultural settings. These particular changes might have occurred due to the selective pressures applied by pesticide treatment. The genetic background and inherent genetic diversity of populations may also influence the variation in mutation sites. Different populations might have unique genetic histories, affecting how mutations develop across different amino acid sites. Additionally, specific mutation sites could rely on the pesticide’s mechanism of action and chemical structure [46].

The *ace1* gene has been targeted explicitly by several pesticides, resulting in distinctive mutational patterns related to resistance. In *P. xylostella* (the Kar-R strain), four amino acid position substitution sites were identified (A298S, G324A, F470L, and T473I) by the present authors (Figure 1). The first report on the connection of mutations in *P. xylostella ace1* to chlorpyrifos resistance was provided by this study. In earlier research, G324A and A298S mutations were found in *P. xylostella* strains that were resistant to acephate [47], prothiofos [13], and mevinphos [48]. The presence of the A298 residue in the oxyanion hole of the recognized ‘FGESAG’ motif around the active serine may aid in stabilizing the tetrahedral structure during catalysis and change the configuration in the active site. It alters sensitivity to the enzyme that inhibits the bond with organophosphate pesticides, resulting in resistance [49,50]. Similarly, the cotton aphid (*Aphis gossypii)* has been seen to develop resistance against pesticides due to the alanine to serine substitution [51,52].

A298S mutations have been identified in the *ace1* gene in the plant bug *Apolygus lucorum* [53]. Sequence analysis of an amino acid mutation in *Chilo suppressalis* indicated a high association between the phenotypic levels of resistance to triazophos organophosphate pesticide and the frequencies of the A298S mutation [54].

Gene expression in mutant *ace* indicates a modest degree of resistance in *Drosophila* (G368A; [55]) and house flies (G365A; [19]). As shown for the G365A mutation in the housefly, the affinity of AChE1 for acetylthiocholine iodide was lower in the resistant population than in the susceptible strain [19]. As anticipated, The G365A mutation in the housefly’s catalytic triad alters the neighboring glutamate structure. The space in the active site may change due to the G324A mutation, which may impact AChE activity.

For the first time, in *P. xylostella ace1*, the mutations T473I and F470L in the Kar-R strain are reported here. Most mutations impacting pesticide entry have been discovered close to the enzyme’s active gorge site. The gorge becomes progressively constricted as a result of substituting a large amino acid with a smaller one. This approach may have facilitated the development of resistance by preventing the pesticide from reaching the active site. Since the structure of the acetylcholine molecule is thinner than that of the majority of chemicals, it is possible to tighten the entrance area, which serves as one of the methods for acquiring resistance in insects [56]. In particular, the A298S and G324A mutations in AChE1 significantly decrease the binding affinity of insecticides.

The results of molecular dynamics simulations of prothiofos-resistant *ace1* demonstrated that the mutant (D132G, A201S, and G227A) *ace1* exhibited little structural deviation compared with the wild-type, indicating structural instability. Furthermore, docking experiments showed that these mutations impair intermolecular hydrogen bonding interactions, affecting both the prothiofos and all pesticide binding [22]. We investigated the impact of four mutations (A298S, G324A, F470L, and T473I) on protein conformational behavior, including structural stability and residual flexibility (Figure 6 and Figure 7). Our molecular dynamics (MDs) results indicated that mutations primarily occurred in the active site. These mutations significantly increased the stability and reduced the residual flexibility of *ace1*, differing from the findings of [22] on *P. xylostella* prothiofos resistance (A201S, G227A), which showed a decrease in stability and an increase in residual flexibility in *ace1*. A comparison of amino acid flexibility between sensitive and resistant proteins revealed greater flexibility in the 425–440 amino acid region (Figure 8), located in the mutant active site ((Glu422) and Acyl pocket), increasing the entropy of the active site. This increased entropy could also explain the decreased enzyme affinity for the inhibitors. The mutations likely altered the orientation of the insecticide hydrophobic tail within the active site, establishing weaker dynamic interactions with the protein. Conformational changes in the AChE1 structure may also change the conformation of the binding site, potentially explaining the resistance of AChE1 to insecticide. These findings support using an RNAi-based pest control strategy as a potential approach for pest management [57].

However, our findings showed that the *ace2* copy ratios of the *P. xylostella* genome are almost equal in both populations. Additionally, *ace1* is more widely expressed and actively transcribed in *P. xylostella* than *ace2*. A higher transcription level of *ace1* remarkably suggests that it encodes a principal and perhaps only detectable *ace* (Figure 3). In contrast to *ace1*, the transcription level of *ace2* was very low. The same result has been reported for *P. xylostella* [8]. Based on the pattern and degree of *ace1* transcription, the Gu-S and Kar-R strains of *ace1* exhibited substantially different expression patterns in our present study. Additionally, the Gu-S strain generated significantly high levels of enzymes [20]. According to earlier studies, the quantity of AChE mRNA in the resistant strains was approximately 1.5 times [58] and 1.07 times [59] greater than in the susceptible strains. This increase may explain why the Kar strain showed resistance to organophosphates in our present study. Moreover, we have seen that chlorpyrifos resistance may develop due to *ace1* mRNA downregulation. In addition, downregulating of ACh receptor(s) results in chlorpyrifos resistance. The downregulation of ACh receptors, which might lead to tolerance to larger doses of organophosphate, may result from long-term suppression of AChE [59].

## 5. Conclusions

This study significantly contributes to understanding chlorpyrifos resistance mechanisms in *P. xylostella* by identifying critical amino acid substitutions in the AChE1 enzyme and analyzing the structural implications of these substitutions by molecular docking and dynamic simulations. These findings provide a solid foundation for designing insecticides to overcome resistance, facilitating sustainable pest management practices. Further research should build upon these insights to develop effective insect pest control strategies while minimizing resistance development.

## Figures and Tables

**Figure 1 insects-15-00144-f001:**
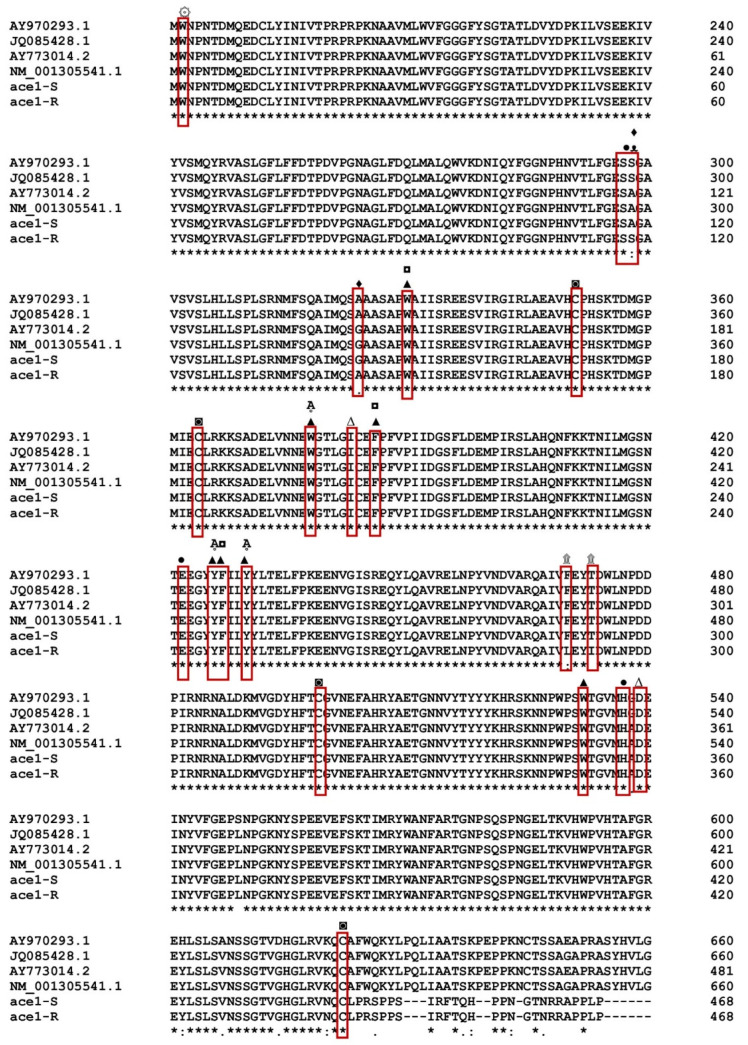
Multiple sequence alignments of *ace1* from Gu-S with Kar-R and other *P. xylostella ace1.* * same amino acide. ۞ Choline-binding site, ● catalytic triad (S200, E327 and H440), ∆ &▲14 aromatic residue in the line *Torpedo Ace* (conserved ▲, nonconserved ∆), ᴥ oxyanion hole, Ḁ acyl pockets, ◘ peripheral anionic subsites, ◙ cysteine residues forming intramolecular disulfide bonds, ۩ new mutation, ♦ reported mutation (AAY34743.1/*P. xylostella*), (AFI47642.1/*P. xylostella*), (AAV65825/*P. xylostella*), (NP_001292470.1/*P. xylostella*), R (Kar-R), and S (Gu-S).

**Figure 2 insects-15-00144-f002:**
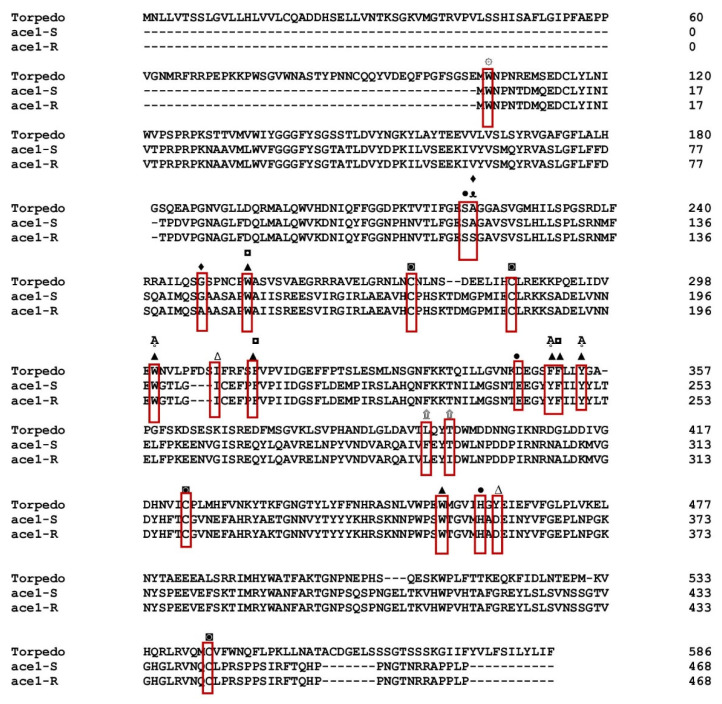
Alignment and comparison of the amino acid sequences of *ace1* from resistant and sensitive with *ace T. californica* populations. (sp|P04058.2/ *T. californica*), R (Kar-R), S (Gu-S). ۞ Choline-binding site, ● catalytic triad (S200, E327 and H440), ∆ and ▲14 aromatic residue (conserved ▲, nonconserved ∆), ᴥ oxyanion hole, Ḁ acyl pockets, ◘ peripheral anionic subsites, ◙ cysteine residues forming intramolecular disulfide bonds, ۩ new mutation, ♦ reported mutation.

**Figure 3 insects-15-00144-f003:**
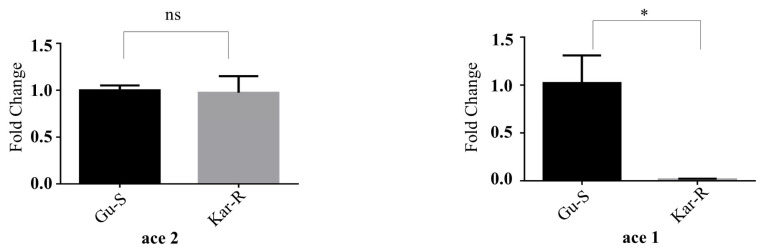
*ace1* and *ace2* gene expression levels in third-instar larvae of *P. xylostella*. An independent *t*-test analyzed differences between treatments, and statistical significance was defined as follows: * *p* < 0.05. ns, nonsignificant (data shown are means ± SEMs).

**Figure 4 insects-15-00144-f004:**
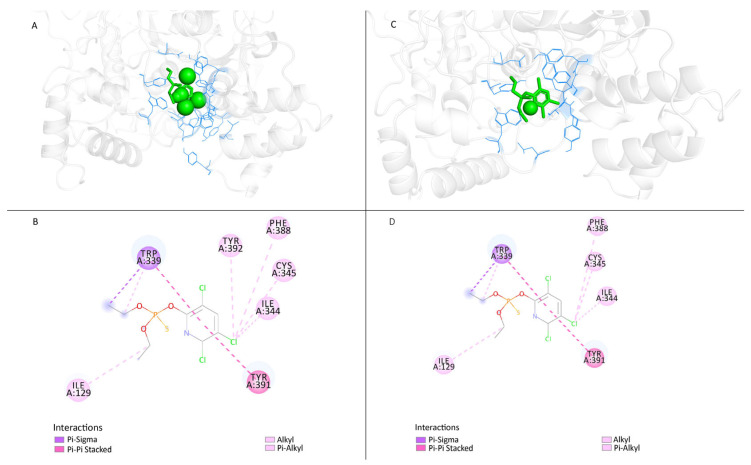
Structures of *ace1* proteins predicted by the SWISS-MODEL server and visualized with Discovery Studio. The white cartoon represents the protein *ace1*, the green spherical sticks represent the small molecule chlorpyriphos oxan (ligand) (2921-88-2), and the blue lines represent the amino acids where the small molecule interacts with the protein. Diverse colors correspond to different interaction modes, with numbers and letters denoting amino acid names and their respective positions engaged in the interaction. (**A**). S-protein (Gu-S) and the ligand; (**B**). Interaction between the s-protein and the ligand; (**C**). Mutant protein (Kar-R) and the ligand; (**D**). Interaction between the mutant protein and the ligand.

**Figure 5 insects-15-00144-f005:**
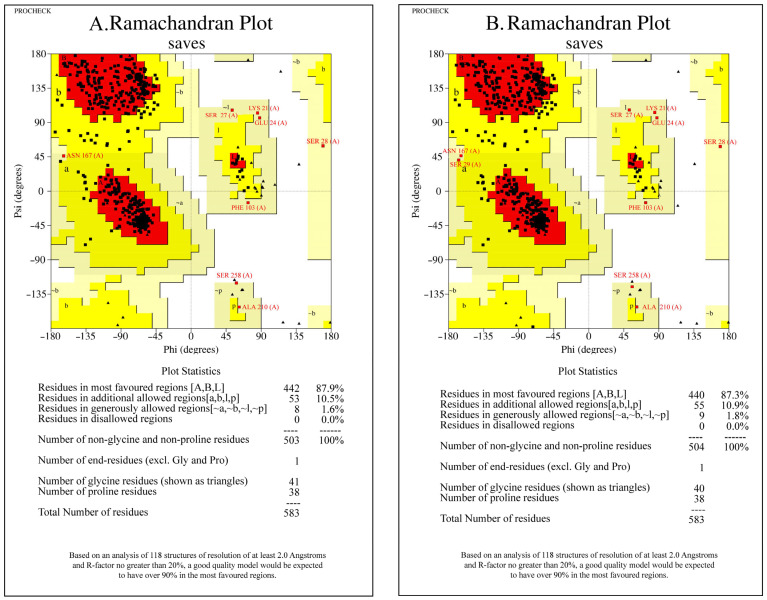
Confirmation of the *ace1* protein in the susceptible (**A**) and resistant (**B**) strains via Ramachandran plots and 3D verification via the SWISS-MODEL. The red, yellow and darkish-yellow areas represent the most favorable, favorable, and disallowed regions. The phi and Psi bonds represent torsion angles, which predict the possible conformations of the peptides.

**Figure 6 insects-15-00144-f006:**
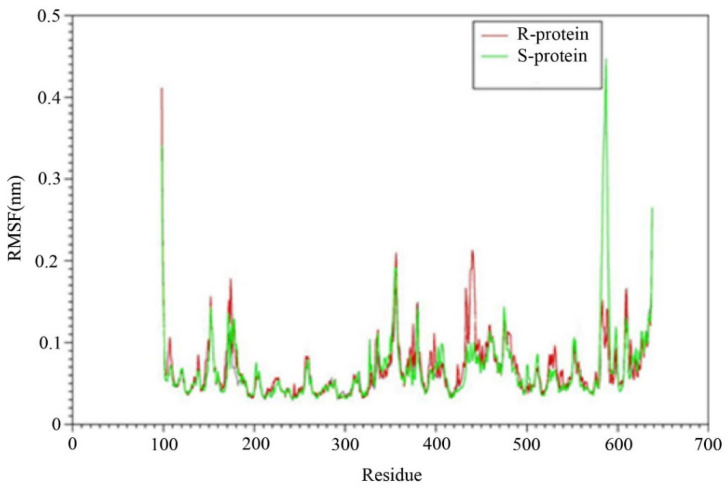
Changes in the root mean square deviation (RMSD) of R (red) and S-protein (green) with respect to the first snapshot during the simulation as a function of time.

**Figure 7 insects-15-00144-f007:**
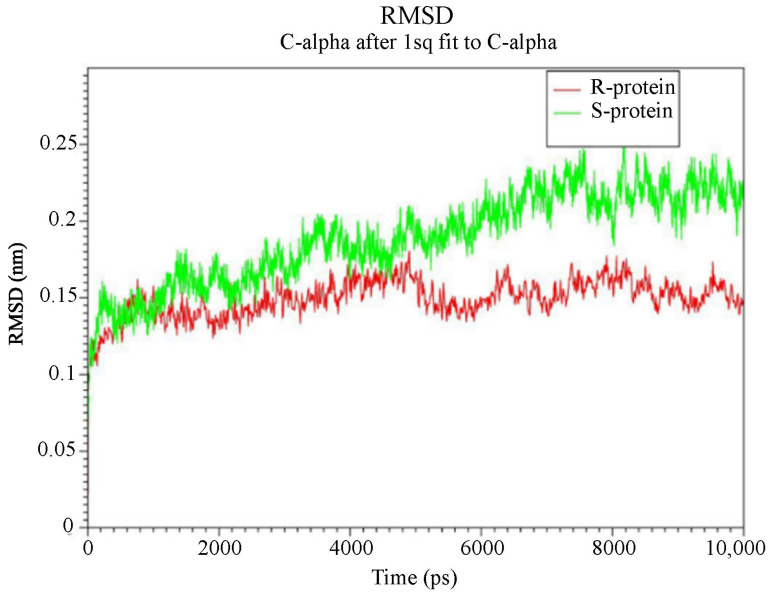
Root mean square fluctuations (RMSFs) of Cα atoms R (red) and S-protein (green) during the simulation as a function of time.

**Figure 8 insects-15-00144-f008:**
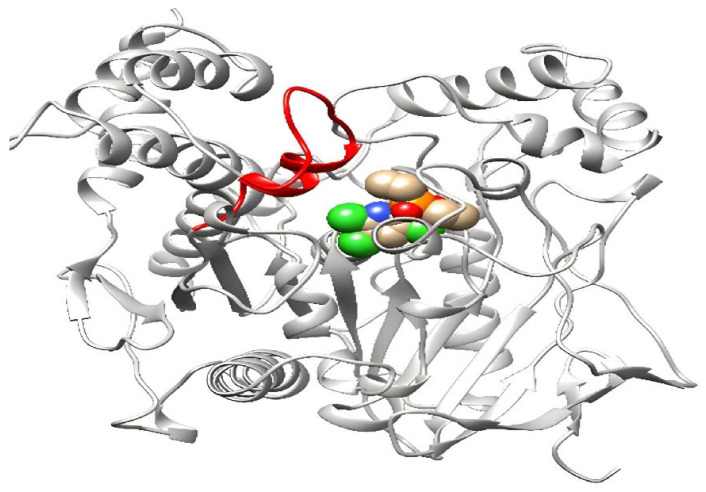
3D visualization of the resistant AChE protein in presence of chlorpyriphos. Here, the area with high flexibility (420–440) is red in AChE. The ligand is also spherical at the junction. The ligand, graphically represented in Ball and stick model with colors.

**Figure 9 insects-15-00144-f009:**
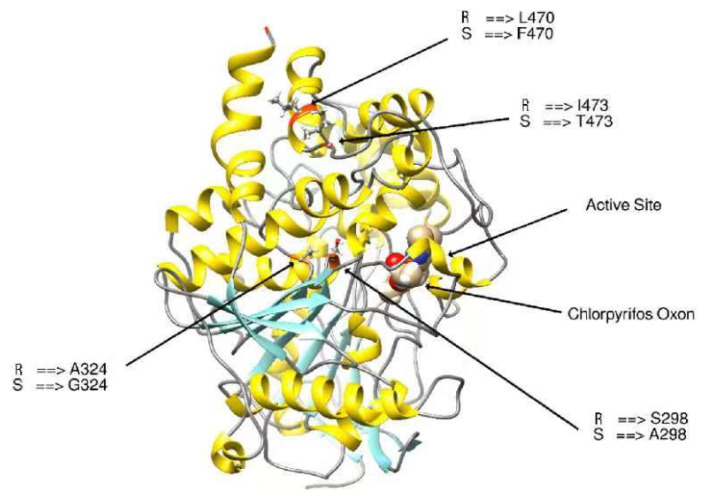
3D visualization of the comparison of amino acid changes in the R (Kar-R) and S (Gu-S) strains of the ACE1 enzyme. The ligand, graphically represented in Ball and stick model with colors.

**Table 1 insects-15-00144-t001:** Sequences of the primers used in this study.

Sequence (5′-3′)	Primer	Amplicon Length (bp)	E ^a^ (%)	R2 ^b^	Purpose
F: CTACAACCCCGAGCTGGACACCATC R: ACACTCGATCATAGCCCATGTCAGTC	ace1-FM2 ace-R2	900	98.4	995	ace1 ORF-PCR ace1 ORF-PCR
F: GACACTCCCGATGTCCCTGGAAAC R: TGGCTGAAACTAACGGCTGCGACG	ace1-F2 ace-RM	1200	98.3	0.999	ace1 ORF-PCR ace1 ORF-PCR
F: TACGCCAAGACCGTGATGGGAG R:CGAAGTAGTTGGTGGGACACACGAAGAAG	ace-FM5 ace-RM	1400	99.3	0.997	ace2 ORF-PCR ace2 ORF-PCR
F: GACACTCCCGATGTCCCTGGAAAC R: ACACTCGATCATAGCCCATGTCAGTC	ace-F2 ace1-R2	300	99.2	0.995	ace1 expression-qPCR
F: AATGCACCGTGGAGTTGGATGACAGG R: TTTTGATTGCTTCCAAGAAGAACTTC	ace2-F2 ace2-R2	250	98.4	0.996	ace2 expression-qPCR
F: GTTGTTGGGAAGTTGACC R: CAGTGCGGCATTCAGT	18SrRNA-F 18SrRNA-R	187	96.9	0.997	Reference gene for ace1 and ace2
F: CCAATTTACCGCCCTACC R: TACCCTGTTGTCAATACCTCT	RPL32-F RPL32-R	168	103.8	0.997	Reference gene for ace1 and ace2

^a^ PCR efficiency ^b^ Regression coefficient.

## Data Availability

The raw data supporting the conclusions of this article will be made available by the authors, without undue reservation.

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
