# Peer review of "Resistance Mechanism of Plutella xylostella (L.) Associated with Amino Acid Substitutions in Acetylcholinesterase-1: Insights from Homology Modeling, Docking and Molecular Dynamic Simulation"

_insects, 2024, doi:10.3390/insects15030144_

Round 1

Reviewer 1 Report

Comments and Suggestions for Authors

Structural protein analysis and proteome analysis of agonist in insect pests are important for understanding the mechanism of resistance evolution. I have some comments.

1.     Authors used the genome of Torpedo california for the comparison with P. xylostella in this study. Why do authors not use the Drosphila genome or Silkworm genome ? Two insect genome is useful for the annotation and genome comparison. Please retry the comparison of genome data using Drosphila or Silkworm genomic data.

2.     Please replace DBH with P. xylostella in a hole manuscript.

3.     Please try the vector format of all figures. Readers cannot read small letters in figure panels and figreus. 

Reviewer 2 Report

Comments and Suggestions for Authors

The article titled "Resistance mechanism of Plutella xylostella (L.) associated with amino acid substitutions in acetylcholinesterase 1: insights from homology modeling, docking, and molecular dynamic simulation" investigates the molecular mechanisms underlying resistance to the insecticide chlorpyrifos in Plutella xylostella, focusing on the ace1 gene. Overall, the article provides valuable insights into the molecular mechanisms of pesticide resistance in this species, with a well-executed methodology and clear presentation of results; however, there is only one part that is not explained in detail, and that is the dose-response bioassays for determine the susceptibility of the two populations of P. xylostella. It is unclear whether the susceptible population is a lab-susceptible strain or a field-susceptible population. If it is "field-susceptible", this population may still be resistant compared to a lab-susceptible strain. Please clarify.

Furthermore, it would be important to explain in detail how the dose-response bioassays were carried out because it is not clear to me whether the bioassays were carried out or came from Zolfaghari et al. (2019)
